

# Coral reefs as a source of climate-active aerosols

Rebecca L. Jackson[1],[*], Albert J. Gabric[2],[*] and Roger Cropp[1]

[1] School of Environment and Science, Griffith University, Gold Coast, QLD, Australia
[2] School of Environment and Science, Griffith University, Nathan, QLD, Australia
[*] These authors contributed equally to this work.

## ABSTRACT

We review the evidence for bio-regulation by coral reefs of local climate through stress-induced emissions of aerosol precursors, such as dimethylsulfide. This is an issue that goes to the core of the coral ecosystem's ability to maintain homeostasis in the face of increasing climate change impacts and other anthropogenic pressures. We examine this through an analysis of data on aerosol emissions by corals of the Great Barrier Reef, Australia. We focus on the relationship with local stressors, such as surface irradiance levels and sea surface temperature, both before and after notable coral bleaching events. We conclude that coral reefs may be able to regulate their exposure to environmental stressors through modification of the optical properties of the atmosphere, however this ability may be impaired as climate change intensifies.

## INTRODUCTION

Coral reefs cover some 600,000 square kilometers of the earth's surface (0.17% of the ocean surface), with coral ecosystems amongst the most diverse on the planet (*Knowlton, 2001*). Coral reefs currently provide a range of ecological services including food and shelter for a variety of marine species, nutrient cycling, as well as income from tourism and fisheries for about 500 million people world-wide (*Moberg & Folke, 1999*; *Hoegh-Guldberg et al., 2007*, *2017*). The total economic value of direct and indirect services that corals provide is estimated to be in the billions to trillions of $US per annum—the highest of all quantified biomes (*Costanza et al., 2014*). However, coral reefs globally have suffered long-term decline in abundance, diversity, and habitat structure due to overfishing and land-based pollution, with most reefs already degraded by the late 19th C (*Pandolfi et al., 2003*). Live coral cover has decreased significantly since baseline monitoring began in the late 1970s, anywhere from 46% to 93%, depending on the region (*Jackson, 2008*), causing many scientists to doubt their long term survival without the use of non-conventional interventions (*Knowlton & Jackson, 2008*; *Anthony et al., 2017*).

On many reefs, reduced stocks of herbivorous fishes together with increased sediment and nutrient loading from land-based activities (*Gabric & Bell, 1993*; *Bell, Elmetri & Lapointe, 2014*) have caused ecological regime shifts away from the original dominance by corals to a preponderance of fleshy seaweed (*Hughes et al., 2007*; *Brodie et al., 2011*). These

Corresponding author
Albert J. Gabric,
a.gabric@griffith.edu.au

regime shifts can occur suddenly (*Bestelmeyer et al., 2011*), and are often irreversible (*Schmitt et al., 2019*). In non-linear systems theory these alternate stable states are known as alternate attractors or basins of attraction (*Walker et al., 2004*). Coral-to-macroalgae regime shifts cause severe changes in a coral ecosystem by altering biotic interactions, disrupting trophic structure, lowering biodiversity, and changing the productivity of reef fisheries (*Hempson et al., 2018*).

Since the late 20th century coral reefs have been subjected to a new range of environmental threats associated with climate change that can seriously jeopardize their continued existence unless radical change occurs in the governance and management of reef systems (*Hoegh-Guldberg & Bruno, 2010*; *Anthony et al., 2017*; *Hughes et al., 2017*). These threats include increasingly frequent and extensive marine heat waves (*Oliver et al., 2018*; *Babcock et al., 2019*), often leading to severe bleaching, and ocean acidification that modifies carbonate chemistry and reef calcification (*Hoegh-Guldberg et al., 2007*). Coral bleaching is caused by the synergistic effect of elevated light and temperature, leading to the breakdown of normal symbiont photosynthetic pathways and causing damage to the host and expulsion of the algal symbionts (*Lesser & Farrell, 2004*). Although mass coral bleaching appears to be a relatively recent phenomenon with reports first emerging in the 1980s (*Glynn, 1983*), the problem has rapidly amplified with mass bleaching events occurring in the late 1990s (*Lough, 2000*) and again during 2015–16, the latter affecting 75% of Indo-Pacific coral reefs, including 84% of Australia's tropical reefs (*Hughes et al., 2018a*).

Interestingly, reduced incoming light due to cloudy conditions has been shown to mitigate bleaching in the Pacific (*Mumby et al., 2001*). Coral reefs within or near the western Pacific warm pool (WPWP)—the so-called "coral triangle"—have had fewer reported bleaching events relative to reefs in other regions (*Kleypas, Danabasoglu & Lough, 2008*). Analysis of sea surface temperature (SST) indicates the warmest parts of the WPWP have warmed less than elsewhere in the tropical oceans, supporting the existence of a thermostat mechanism that depresses warming beyond certain temperature thresholds. One of the suggested thermostat mechanisms was via a cloud-SST feedback (*Ramanathan & Collins, 1991*), with a more detailed description of cloud feedbacks given by *Stephens (2005)*.

Over the past 10–15 years, field and laboratory studies have provided evidence for the existence of a hitherto unrecognized climate bio-regulation process in coral reefs (*Broadbent, Jones & Jones, 2002*; *Broadbent & Jones, 2004*; *Jones, 2015*). This is through the production of a suite of volatile compounds that can act as precursors of marine biogenic aerosol (MBA) in response to physiological stress experienced by the coral related to high irradiance or ocean temperature. In remote marine atmospheres, these secondary biogenic aerosols are thought to influence the local radiative budget through backscattering of incoming short-wave solar radiation, and indirectly through their effect on cloud microphysics and precipitation forming processes. The climate regulation potential of MBA was first discussed over 30 years ago by *Charlson et al. (1987)*, with the so-called "CLAW hypothesis" spawning a plethora of related research regarding the possibility of a natural thermostat which would offset the warming caused by

anthropogenic greenhouse gases (GHG). This research theme has proved to be remarkably resilient and continues to the present day (*Gabric et al., 2018*; *Mahmood et al., 2019*), although the strength and sign of any MBA feedback on climate warming is likely to be regionally variable and is still uncertain at the global scale (*Ayers & Cainey, 2007*; *Heinze et al., 2019*).

Our understanding of aerosol-climate interactions although growing, is still incomplete, being identified by the Intergovernmental Panel on Climate Change (IPCC) as one of the key sources of uncertainty in our knowledge of Earth's energy budget and anthropogenic climate forcing (*Schneider et al., 2017*; *Simpkins, 2018*). This is particularly true of coral reef ecosystems where the relationship between MBA emissions and coral physiological stress is complex (*Jackson, Gabric & Cropp, 2018*). Here we review the current knowledge of MBA emissions from coral reef ecosystems, examine changes in aerosol emissions as a response to physiological stressors and discuss the implications for the future resilience of coral reefs in response to climate change related stressors.

## SURVEY METHODOLOGY

The quite separate fields of coralline ecology and aerosol-climate interactions both have a long and rich history. Unsurprisingly however, given the distinct disciplines involved, the intersection of these fields is relatively recent. Indeed, most of the published research on biogenic aerosol emissions by coral reefs has appeared in the last two decades. Notwithstanding the emerging nature of the field, there are numerous aspects of the topic that bridge the disciplines of climatology, aerosol science and coral reef ecology. Consequently, we have chosen to employ an integrative or critical review approach. Our aim is to assess the current evidence for coral reef bio-regulation of climate and to synthesize the literature in a way that will enable a new theoretical framework and paradigm to emerge (*Torraco, 2005*, *2016*). Literature searches were conducted using the key bibliographic databases both full text, such as Web of Science and Google Scholar and Abstract only databases, such as Scopus. The search time frame was limited to the last 30 years as most of the relevant literature has been published relatively recently. Boolean searches were used to narrow the results to capture the literature on both coral reefs and aerosols. Search terms such as "coral stress", "aerosol emissions", "dimethylsulfide AND corals" were used.

### Marine biogenic aerosol

Aerosols are minute solid or liquid particles suspended in the atmosphere and are derived from a variety of natural and anthropogenic sources, ranging from industrial processes, volcanic eruptions, biomass burning and marine ecological processes. Aerosol particles are either emitted directly to the atmosphere (primary aerosols) or produced in the atmosphere from precursor gases (secondary aerosol). All atmospheric aerosols scatter incoming solar radiation, and a few aerosol types (e.g., black carbon) can also absorb solar radiation. Aerosols that mainly scatter solar radiation have a cooling effect, by enhancing the total reflected solar radiation from the Earth (*Twomey, 1977*). However, it is the interaction of some aerosols with clouds that leads to a suite of complex but radiatively

important effects. The concentration of droplets in clouds that influences planetary albedo is sensitive to the availability of aerosol particles on which the droplets form. An impact on cloud droplet numbers affects rain formation, and thus the cooling effect may be further enhanced by suppressed precipitation followed by increased cloud lifetime, cloud amount and cloud extent (*Albrecht, 1989*; *Pincus & Baker, 1994*). However, notwithstanding recent progress in our understanding of aerosol-climate interactions, there is still uncertainty about the links between microphysical and larger scale mechanisms, and how climate feedbacks may be affected (*Fan et al., 2016*; *Brooks & Thornton, 2018*).

The most convincing evidence for aerosol modulation of cloud properties has been seen in the marine atmosphere (*Hegg, 1999*; *Hegg et al., 2004*), specifically the increase in albedo of marine stratocumulus clouds, which cover about a third of the global oceans. Over the last two decades, the availability of satellite-based data has enabled a better understanding of MBA, which has been shown to play an important role in the radiative budget of remote marine atmospheres and potentially shaping regional climate (*McCoy et al., 2015*; *Fan et al., 2016*; *Vergara-Temprado et al., 2018*). However, despite over three decades of research, there are still gaps in our understanding of the effect of aerosol–cloud interactions on climate (*Ayers & Cainey, 2007*; *Carslaw et al., 2013*). MBA can be primary aerosols consisting of sea-salt and particulate organic matter (*Leck & Bigg, 2005*; *Orellana et al., 2011*; *Modini et al., 2015*), or secondary aerosols formed through the atmospheric oxidation of volatile precursor compounds, such as dimethylsulfide (DMS) (*Andreae & Crutzen, 1997*), organo-halogens (*O'Dowd et al., 2002*) and other organic compounds. In the original CLAW hypothesis MBA precursor compounds such as DMS were thought to be synthesized solely by pelagic phytoplankton, but as shown in Fig. 1, other organisms such as corals and benthic algae are also known to be sources (*Broadbent & Jones, 2004*; *Raina et al., 2013*; *Burdett, Hatton & Kamenos, 2015*). It is now recognized that the synthesis and emission of these biogenic climate active compounds is shaped by a range of marine ecosystem processes (*Liss et al., 2000*; *Carslaw et al., 2010*). Thus, the sea-to-air flux of these aerosol precursor compounds and particles depends in a complex fashion on the structure and dynamics of the entire marine food web (*Simó, 2001*).

### Potential to regulate climate

The effect of an change in atmospheric aerosol concentrations on the distribution and radiative properties of Earth's clouds is the most uncertain component in model projections of the global radiative forcing of climate (*Seinfeld et al., 2016*). This makes it imperative to investigate the current and future sources of these climate-active compounds. However, there are several factors that constrain an improved estimate of the effect of aerosol–cloud interactions. Although aerosol–cloud processes are reasonably well understood at the scale of a single cloud, the difference in scale between the spatial resolution of general circulation models (GCMs) and individual cloud processes introduces considerable uncertainties (*Seinfeld et al., 2016*). Secondly, the change in future aerosol emissions is uncertain, with anthropogenic emission trends already negative in

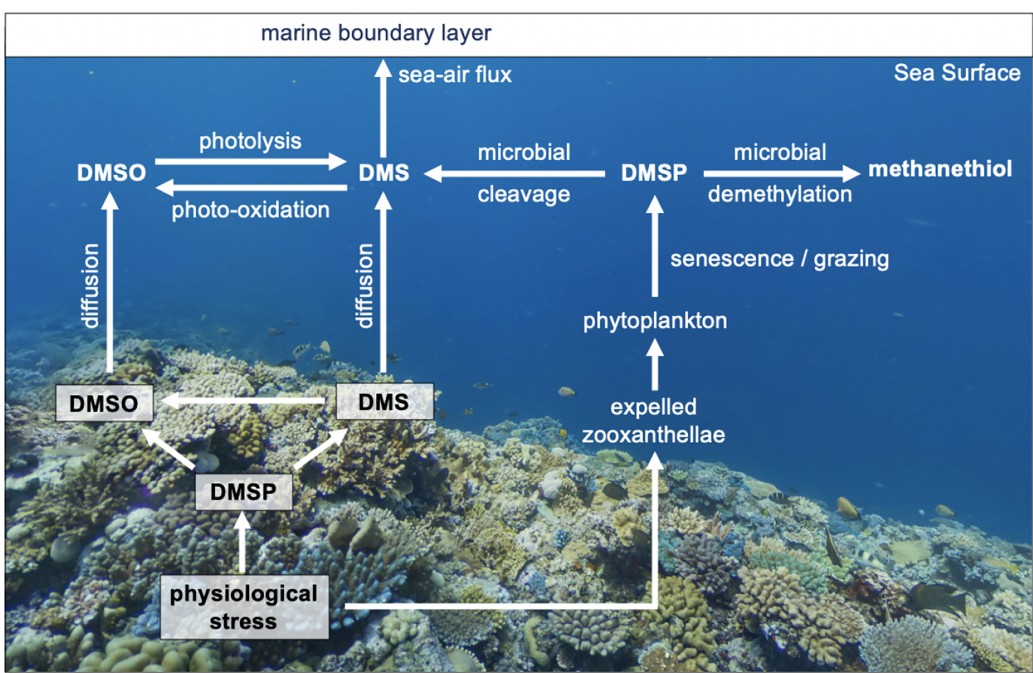

**Figure 1 The cycling of reduced sulfur compounds in coral reef waters.** Simplified overview of the cycling of reduced sulfur compounds in coral reef seawaters and their role in coral homeostasis. Corals upregulate dimethylsulfide (DMSP) biosynthesis and cleavage to dimethylsulfide (DMS) in response to physiological stress. DMS(P) scavenge reactive oxygen to mitigate oxidative damage, forming dimethyl sulfoxide (DMSO).

developed economies (*Zhao et al., 2017*). This trend is likely to be reinforced as pollution controls are implemented in developing economies which are suffering serious impacts from particulate air pollution (*Huang et al., 2014*; *Samset et al., 2018*).

The role of natural aerosol emissions in cloud radiative forcing is less certain but thought to be even greater than that due to anthropogenic aerosol (*Carslaw et al., 2013*). With respect to future trends in natural aerosol emissions such as sulfur-containing MBA, the model projections vary markedly depending on the ocean region considered and the model complexity and parametrizations used (*Gabric et al., 2004*, *2005*; *Cameron-Smith et al., 2011*; *Six et al., 2013*; *Menzo et al., 2018*). In the rapidly warming Arctic Ocean decadal data on DMS suggests a positive trend in emissions (*Galí et al., 2019*). In other parts of the global ocean the trend is not clear, often due to sparseness of the existing database. This is particularly true for the Southern Ocean where the sparse databases are currently being augmented and improved (*Jarníková & Tortell, 2016*; *Webb et al., 2019*). Modelling experiments suggest that increasing atmospheric greenhouse gas concentrations may enhance future DMS emissions, and thus sulfate aerosol concentrations, in both the Arctic and Southern Oceans, introducing a negative feedback to offset the warming (*Gabric, Whetton & Cropp, 2001*; *Gabric et al., 2005*; *Qu et al., 2015*; *Kim et al., 2018*). Increasing ocean acidification may also impact marine DMS emissions, although the sign of the feedback appears to be regionally variable (*Wingenter et al., 2007*; *Archer et al., 2013*, *2018*; *Six et al., 2013*).

### Aerosol precursors in Coral Reefs

Reef-building corals are prolific producers of dimethylsulfoniopropionate (DMSP), a central molecule in the marine sulfur cycle and precursor of DMS (*Broadbent & Jones, 2004*). Both DMS and DMSP are particularly abundant in coral reef ecosystems (*Jones, Curran & Broadbent, 1994*; *Hill, Dacey & Krupp, 1995*), being present in macroalgae (*Broadbent, Jones & Jones, 2002*), coralline algae (*Burdett, Hatton & Kamenos, 2015*), soft corals (*Haydon, Seymour & Suggett, 2018*) and also detected in coral polyps themselves (*Raina et al., 2013*). DMSP is produced by both the algal endosymbiont *Symbiodinium* (*Hill, Dacey & Krupp, 1995*) and coral host (*Raina et al., 2013*) which, together with the breakdown products DMS and dimethyl sulfoxide (DMSO), has various roles in coral reef ecosystems, including oxidative stress protection (*Deschaseaux, Jones & Swan, 2016*; *Gardner et al., 2017*). It has become increasingly clear that the whole coral holobiont (comprised of the coral animal and its associated microorganisms consisting of bacteria, fungi, viruses, and protists including the dinoflagellate algae *Symbiodinium*) is to some degree involved in the synthesis and cycling of these sulfur compounds (*Raina et al., 2010*).

The first hint of a link between coral physiological stress and DMS(P) was noted some decades ago in the Florida keys, where extremely high concentrations of atmospheric DMS were observed after aerial exposure of the reef at low tide (*Andreae, Barnard & Ammons, 1983*), and later a possible effect of stress-related DMS emissions on the local reef climate was also hypothesized (*Hill, Dacey & Krupp, 1995*). A detailed treatment of the anti-oxidant role of dimethylated sulfur compounds was first reported for pelagic phytoplankton by *Sunda et al. (2002)*. More recently this has been extended to other marine organisms such as benthic algae (*Burdett, Hatton & Kamenos, 2015*) and corals themselves (*Deschaseaux et al., 2014b*). The anti-oxidant role of DMSP is especially evident in *Acropora* corals (*Gardner et al., 2016*), the dominant species throughout the Great Barrier Reef (GBR), Australia. *Acropora* are among the highest producers of DMS and increased emissions have been detected in response to increases in sea temperature, solar irradiance and osmotic stress (*Fischer & Jones, 2012*; *Swan et al., 2017*). Seasonal increases in DMS emissions from coral reefs have been observed during low tides when the reef can be aerially exposed (*Hopkins et al., 2016*; *Jones et al., 2018*). If aerial exposure coincides with high irradiance then significant coral mortality can occur (*Anthony & Kerswell, 2007*). Corals can also be stressed during periods of high rainfall when hyposalinity may affect coral physiology (*Gardner et al., 2016*; *Aguilar et al., 2017*).

Notwithstanding the recent progress in the field, there is limited understanding of the mechanisms of DMS production by the coral holobiont and relatively sparse data on either dissolved or atmospheric DMS concentrations in coral reef areas. Similarly, estimates of DMS fluxes to the atmosphere from reefal environments are as yet poorly constrained and not included in global DMS data bases such as that of *Lana et al. (2011)*.

## Effects on local climate

Although the nexus between MBA emissions and changes in the properties of maritime clouds has been debated for a long time (*Ayers & Cainey, 2007*), significant progress in the
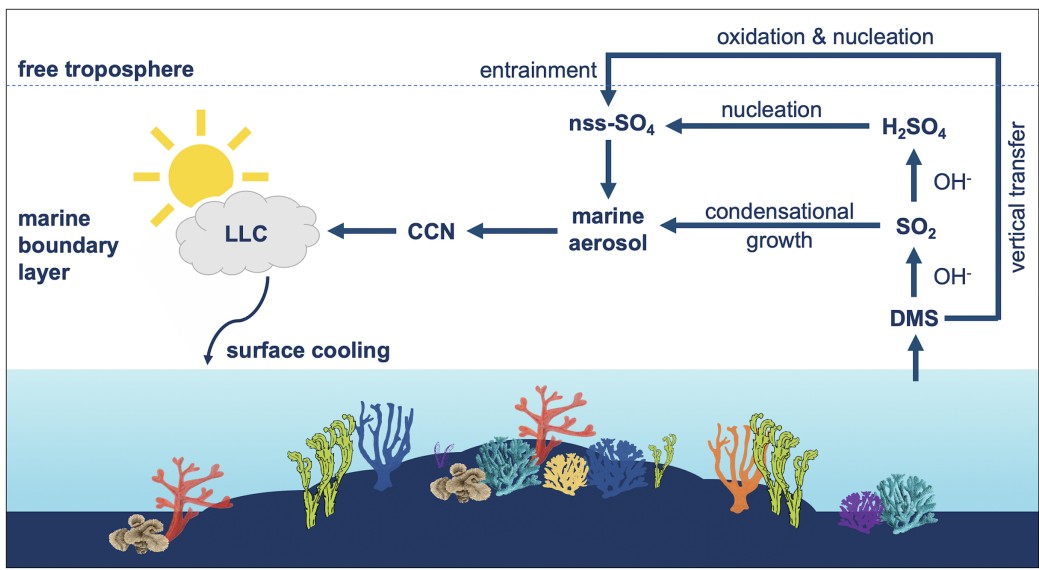

**Figure 2 Influence of dimethylsulfide (DMS) emissions on marine atmosphere over coral reefs.** Summary of the influence of dimethylsulfide (DMS) emissions on marine aerosol, cloud condensation nuclei (CCN) and low-level clouds (LLC) over coral reefs. DMS is oxidized by hydroxyl radicals (OH) to sulfur dioxide ($SO_2$) and secondarily to sulfuric acid ($H_2SO_4$). DMS-derived sulfates may condense onto pre-existing aerosols or undergo homogenous nucleation to form new non-sea salt sulfate (nns-$SO_4$) particles, which can influence the local radiative balance.

understanding of aerosol chemistry and climate has been made in the last two decades (*McNeill, 2017*). The advent of high-resolution satellite data has permitted the analysis of aerosol–cloud interactions over large swaths of the global ocean. Several studies have shown a strong correlation between MBA and marine cloud cover and cloud properties (*Kruger & Grassl, 2011*; *Lana et al., 2012*; *McCoy et al., 2015*). Notwithstanding this, a recent comprehensive review concluded that the relationship between marine biogeochemical processes and cloud formation is potentially significant but still poorly defined (*Brooks & Thornton, 2018*).

In pristine coral reefs such as the WPWP, DMS emissions are thought to be the key driver behind an ocean thermostat which suppresses ocean warming below coral thermal tolerance thresholds (~30 °C) through a build-up of low-level clouds (LLC), as shown in Fig. 2. Despite corals in the WPWP living close to their thermal maxima, few coral bleaching events have been recorded in this region, and although uncertain (due to the possible under-reporting of bleaching events) this resilience to heat stress is thought to be due to cloudiness (*Kleypas, Danabasoglu & Lough, 2008*; *Kleypas et al., 2015*). The role of cloud cover in moderating the intensity of bleaching in the Society Islands was also noted by *Mumby et al. (2001)*. In the GBR, a decadal analysis of the connection between bleaching and solar radiation showed that the area of maximum bleaching corresponded closely to the area of maximum solar insolation (*Masiri, Nunez & Weller, 2008*). Some evidence points to a similar aerosol-climate feedback mechanism operating in the GBR, where although ocean temperatures in north-eastern Australia are warming, SSTs in

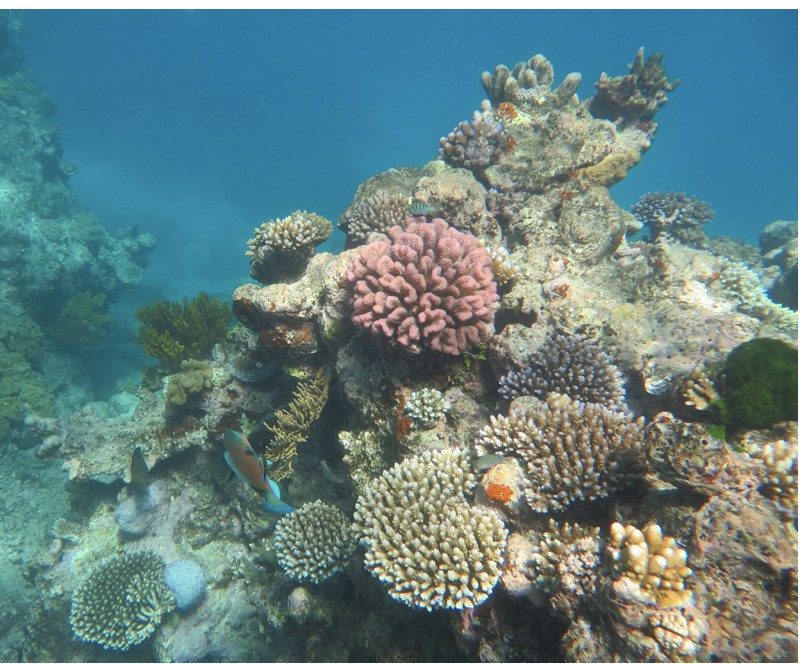

**Figure 3 Diverse coral community at Norman Reef in the northern Great Barrier Reef, Australia.**
Source: Rebecca Jackson.

the northern GBR are rising at a slower rate compared to southern regions (*Lough, 2008*).
As conjectured for the WPWP, this may be due to the high biomass of DMS(P)-producing
corals and the accumulation of DMS-rich air in the prevailing south-east trade winds
over the GBR (*Jones & Trevena, 2005*; *Jones et al., 2017*).

An 18-year time series study (*Jackson, Gabric & Cropp, 2018*) of satellite-derived
fine-mode aerosol optical depth (AOD) in the GBR found that AOD correlated positively
with SST and irradiance and increased two-fold during spring and summer. *Jackson,
Gabric & Cropp (2018)* posit that the positive correlation between AOD and both SST and
irradiance is consistent with enhanced DMS-derived particle formation over the GBR.

## Case study: the GBR, Australia

The GBR is the world's largest coral reef ecosystem, consisting of 3,000 individual coral
reefs spanning 2,300 km of the north-eastern Australian coastline (10°S–24°S) (See Fig. 3).
The Great Barrier Reef Marine Park (GBRMP) was established in 1975 to manage a
diversity of species, including more than 400 Scleractinian (stony) corals, 1,500 species
of fish, 30 species of marine mammals and six of the seven species of marine turtles.
The diversity and size of the GBR makes it incredibly important for tourism, fisheries,
ecosystem services (reviewed in *Stoeckl et al. (2011)*). *Acropora* spp. are the dominant coral
genus throughout the GBR and Indo-Pacific and are strong producers of climatically active
dimethylated sulfur compounds (*Raina et al., 2013*; *Swan et al., 2016*). The GBR and
lagoon waters (347,000 km$^2$) are estimated to emit 20 Gg S year$^{-1}$ as DMS (*Jones et al.,
2018*) and may therefore play an important role in local climate.

**Table 1 Range of atmospheric dimethylsulfide (DMS$_a$) concentrations during field surveys of various ocean regions.**

| Region | DMS$_a$ (nmol m$^{-3}$) | Reference(s) |
|---|---|---|
| Great Barrier Reef, Australia[*] | 0.1–45.9 | *Broadbent & Jones (2006)*; *Jones & Trevena (2005)*; *Jones et al. (2007)*; *Swan et al. (2016, 2017)* |
| North Coral Sea[*] | 0.3–6.9 | *Jones & Trevena (2005)* |
| Gulf of Papua[*] | 0.5–2.9 | *Jones & Trevena (2005)* |
| Bismarck Sea[*] | 1–5.3 | *Jones & Trevena (2005)* |
| Solomon Sea[*] | 1.2–5.3 | *Jones & Trevena (2005)* |
| Bahamas[*] | 0.08–10.8 | *Andreae et al. (1985)* |
| Tropical East Pacific Ocean[+] | 1.1–6.7 | *Andreae et al. (1985)* |
| Indian Ocean[+] | 1.3–11.3 | *Bandy et al. (1996)*; *Conley et al. (2009)*; *Nguyen, Mihalopoulos & Belviso (1990)* |
| North Pacific Ocean | 2.5–11.1 | *Aranami & Tsunogai (2004)* |
| North Atlantic Ocean | 0.03–6.6 | *Andreae et al. (1985)*; *Andreae et al. (2003)* |
| Arctic Ocean | 0.04–47.2 | *Ferek et al. (1995)*; *Lundén, Svensson & Leck (2007)*; *Mungall et al. (2016)*; *Park et al., (2013)* |
| Southern Ocean | 0.5–17.2 | *Andreae et al. (1985)*; *Berresheim et al. (1990)*; *Curran, Jones & Burton (1998)*; *Curran & Jones (2000)*; *Yang et al. (2011)* |
| Mediterranean | 0.3–8.9 | *Kouvarakis & Mihalopoulos (2002)* |
| Sargasso Sea | 0.02–16.3 | *Andreae et al. (1985)* |

**Notes:**
[*] Coral reef dense regions.
[+] Oceanic cruise tracks with intermittent coral reef regions.

The climate of the GBR ranges from sub-equatorial in the north, to sub-tropical in the south, with hot monsoonal summers (November–April) and dry, mild winters (May–October). Corals upregulate the biosynthesis of DMSP and catabolism to DMS during physiological stress caused by exposure to elevated SST and solar irradiance, or due to low salinity associated with seasonal rainfall and riverine discharge (*Raina et al., 2013*; *Deschaseaux et al., 2014b*; *Jones et al., 2014*). These processes likely drive seasonal increases in DMS emissions from the GBR during spring and summer (*Broadbent & Jones, 2006*; *Jones et al., 2018*). Seasonally aberrant spikes in atmospheric DMS (DMS$_a$) also occur when the coral reef is aerially exposed at low tide (*Hopkins et al., 2016*; *Swan et al., 2017*), with atmospheric concentrations reaching 45 nmol m$^{-3}$ (Table 1) over corals at Heron Island in the southern GBR (*Swan et al., 2017*). Similar to the CLAW hypothesis, it is possible that coral reef emissions of DMS may influence the chemical and physical properties of aerosols and cloud condensation nuclei (CCN), thereby increasing the radiative cooling effect of low-level marine clouds over coral reefs. Whether this effect exerts a regional climate impact depends on a number of factors. Any local climate feedback depends on the spatial extent of the reef system, and the strength and direction of prevailing winds which could transport aerosols and their precursor compounds away from the reefal source area (*Cropp et al., 2018*). The large spatial extent of the GBR makes it the most likely candidate for a local aerosol-climate feedback.

## Aerosol climatology of the GBR

Prevailing south-easterly winds accumulate marine aerosols as air is advected northward along the GBR. These aerosols largely consist of clusters of sea-salt, organics (*Mallet et al., 2016*) and non-sea salt sulfates derived from DMS (*Modini et al., 2009*; *Swan et al., 2016*). Aerosol emissions vary seasonally and meridionally in the GBR (*Jackson, Gabric & Cropp, 2018*). In the northern GBR, aerosol loading is highest during the winter dry season and early spring (*Jackson et al., 2020*), coinciding with frequent biomass burning and wildfires on the Australian continent (*Harris et al., 2008*). Here, the situation is complex as the marine biogenic source of aerosol is likely augmented by continental aerosol sources, with quite different composition (carbonaceous) and radiative properties (*Langmann et al., 2009*). Dust storms from the arid inland regions of the continent can also traverse the GBR region later in spring, but these events are episodic and their likelihood of occurrence is low (*Cropp et al., 2013*). Conversely, in the southern GBR (>15°S), seasonal, high frequency increases in aerosol occur in spring and summer, followed by a decline in winter (*Cropp et al., 2018*; *Jackson, Gabric & Cropp, 2018*). This seasonal cycle is commonly observed in the remote MBL and is driven by temperature and irradiance-dependent shifts in ocean biology (*Korhonen et al., 2008*; *McCoy et al., 2015*; *Gabric et al., 2018*). Seasonal peaks in phytoplankton biomass are usually found during the summer wet season when fluvial nutrient loads to the inshore GBR are high (*Gabric, Hoffenberg & Boughton, 1990*; *Brodie et al., 2007*). Given the remote location and vast size of the GBR, seasonal shifts in stress-induced emissions of DMS from corals, and other volatile organics such as isoprene from reef sediments (*Swan et al., 2016*; *Hrebien et al., 2020*), may be driving the increase in aerosol in spring and summer.

Several studies support the hypothesis that the GBR is a significant source of marine aerosols. Early field studies found that total atmospheric particle concentration was up to seven times higher in maritime air directly over the GBR compared to the adjacent open ocean (*Bigg & Turvey, 1978*). Three decades later, observations of nucleation events at Lizard Island in the northern GBR identified a strong seasonal cycle in atmospheric particle concentration (*Leck & Bigg, 2008*). The concentration of nucleation mode aerosol was an order of magnitude higher in spring, reaching up to 40,000 $cm^{-3}$, compared to 4,000 $cm^{-3}$ in winter (*Leck & Bigg, 2008*), following a similar seasonal cycle to that of DMS emissions from the GBR (*Broadbent & Jones, 2006*; *Jones et al., 2018*). Other recent field studies have observed nucleation events in the southern GBR during daylight, low relative humidity (~60%) and low wind speeds (*Modini et al., 2009*; *Swan et al., 2016*), when conditions for the local gas-phase nucleation of DMS-derived sulfates is favorable (*Chang et al., 2011*). New fine-mode aerosols (≤1 μm) consisted of ~40–50% organics and 50–60% sulfates, were likely derived from DMS emissions from the coral reef. Remote sensing approaches have also demonstrated a significant correlation between estimates of coral physiological stress and aerosol optical depth (AOD) in the GBR, especially when wind speeds are low allowing for longer aerosol residence time over the reef area (*Cropp et al., 2018*; *Jackson, Gabric & Cropp, 2018*).

## Recent coral bleaching events

Corals in the GBR are exposed to multiple stressors, including ocean warming, acidification and poor water quality, which individually and synergistically diminish coral resilience and can result in coral bleaching and subsequent mortality (*Anon, 2019*). Reduced cloud cover and marine heatwaves often coincide with an El Niño phase of the Southern Oscillation Index and are the most common drivers of bleaching events in the GBR (*McGowan & Theobald, 2017*; *Hughes et al., 2018b*). Over the past two decades, the GBR has experienced five mass thermal bleaching events in the summers of 1997–1998, 2001–2002, 2005–2006, 2015–2016 and 2016–2017. Inshore coral reefs are particularly vulnerable to declining water quality due to runoff from adjacent catchments and urban areas (*Gabric & Bell, 1993*; *Brodie et al., 2011*; *De'ath et al., 2012*). In the summers of 2008–2009 and 2010–2011, La Nina associated flooding, low salinity and eutrophication combined to result in mass coral bleaching (*Thompson et al., 2011*, *2013*). These stressors are often exacerbated by destructive wave action and hyposalinity resulting from fluvial inputs or rainfall associated with tropical cyclones, which frequently occur in summer (*Anon, 2011*; *De'ath et al., 2012*).

*Acropora* spp. are temperature-sensitive and are particularly vulnerable to rises in SST (*Fischer & Jones, 2012*; *Hughes et al., 2018a*). The two most recent mass coral bleaching events occurred due to marine heatwaves in the summers of 2015–2016 and 2016–2017. SST was well above average in the summer of 2015–2016 (*Jackson, Gabric & Cropp, 2018*), resulting in wide spread coral bleaching and mortality (*Hughes et al., 2018a*). This was the worst coral bleaching event on record in the GBR, affecting 92% of coral reefs in the marine park (*Anon, 2017*), with *Acropora* spp. suffering catastrophic mortality (*Hughes et al., 2018a*). Field surveys conducted by the Australian Institute of Marine Science (AIMS) estimated that 29% of shallow-water corals were lost reef-wide, with the largest loss reported in the far northern GBR (~75%) (*Anon, 2017*). Corals in the southern GBR were least affected by this event as SST rapidly subsided with category five tropical cyclone Winston in late February, although temperature-sensitive *Acropora* and *Pocillopora* colonies were still affected (*Kennedy, Ordoñez & Diaz-Pulido, 2018*).

Sea surface temperature remained above average throughout the GBR in winter 2016 and by the following summer, resulted in a second mass coral bleaching event. Coral mortality was lower in the far northern GBR during this event due to the loss of many temperature sensitive corals during the previous summer (*Anon, 2018*). Consequently, the most severely affected region shifted south to the north-central GBR in 2017. Temperature-sensitive spawning corals comprise ~90% of reef-building corals in the GBR, many of which were lost during these consecutive coral bleaching events. Consequently, larval recruitment has fallen by an average of 89% across the GBR (*Hughes et al., 2019*) resulting in regional scale shifts in community structure (*Hughes et al., 2018b*).

### Changes in aerosol emissions before and after bleaching events

Corals in the GBR increase DMSP biosynthesis and catabolism to DMS in response to oxidative stress (*Deschaseaux et al., 2014b*). Reactive oxygen species (ROS) are released from zooxanthellae photosystems when damage caused by intense photosynthetically

active radiation (PAR) or elevated SST exceeds photoprotective mechanisms (*Lesser et al., 1990*; *Yakovleva et al., 2009*). DMSP and particularly DMS, have a high affinity for these ROS (*Sunda et al., 2002*), and act as an efficient antioxidant system in corals to help protect against stressors leading to coral bleaching. When oxidative stress exceeds coral's photoprotective mechanisms, the rate of DMS(P) oxidation to DMSO increases, and ambient DMS concentrations decline (*Fischer & Jones, 2012*; *Deschaseaux et al., 2014b*). When *Acropora* spp. in the southern GBR were exposed to SST $\geq 26\,^{\circ}\text{C}$ or PAR $\geq 6\,\text{mol m}^{-2}\,\text{h}^{-1}$, DMS emissions declined by 93% and 82%, respectively (*Fischer & Jones, 2012*). A decline in DMS emissions results in fewer aerosol precursor compounds and potentially less aerosol formation events and condensational growth of pre-existing aerosols above the coral reef.

The non-linearity in coral physiological stress and potential effects on aerosol loading was recently investigated during four mass thermal coral bleaching events between 2001–2017 (*Jackson, Gabric & Cropp, 2018*). The coherence between satellite-derived anomalies of fine-mode (<0.1 μm) AOD and estimates of coral thermal stress, calculated as degree-heating weeks (DHW), was examined. Prior to coral bleaching, SST increased, and corals were likely emitting large quantities of DMS in an attempt to mitigate thermal stress (*Raina et al., 2013*). During this time, AOD was highly variable and often above the long-term average (2000–2017). However, the pattern of DMS emissions with coral physiological stress is non-linear and shows a decline when the coral thermal stress threshold is exceeded (*Fischer & Jones, 2012*). This threshold or tipping-point was calculated as the climatological maximum summertime SST and ranged from 27.3 °C at Heron Island in the southern GBR, to 29.1 °C at Fife Island in the far northern GBR (*Jackson, Gabric & Cropp, 2018*). As SST approached this tipping-point, and DHW and field-based reports indicated that coral bleaching was occurring, AOD declined to average, or below average levels where coral bleaching and mortality was severe (*Jackson, Gabric & Cropp, 2018*). The synchronous decline in AOD with the onset of coral bleaching may have been driven by a decline in DMS and MBA emissions from the coral reef. Although the AOD can be affected by a range of aerosol types, the spatio-temporal coherence between the timing of coral bleaching and sharp AOD changes support the hypothesis of a strong causal link between coral physiological stress and aerosol emissions in the GBR.

### Implications for future Coral Reef resilience and adaptation

Reef-scale micrometeorology is an important determinant of the extent and severity of coral bleaching in the GBR (*McGowan & Theobald, 2017*; *McGowan et al., 2019*). DMS and other volatile biogenic compounds influence aerosol and cloud properties in the remote MBL (*Gabric et al., 2013*; *Fiddes et al., 2018*; *Sanchez et al., 2018*) and likely play an important role in the local climate of the GBR. However, ongoing coral reef degradation and bleaching could lead to a decline in DMS emissions from the GBR, with concerning implications for coral resilience to future temperature rises. A decline in biogenic aerosol emissions could weaken the aerosol and LLC radiative cooling effect in the GBR,

exacerbating coral physiological stress and potentially leading to more frequent bleaching events.

Coral DMS(P) biosynthesis increases with thermal and irradiance stress, followed by oxidation by ROS to DMSO in temperature-sensitive species. These species are the dominant reef-building corals in the GBR and are the strongest individual producers of DMS(P) (*Swan et al., 2016*). The rate of oxidation to DMSO determines the amount of DMS available to be ventilated to the MBL. Thus when oxidative stress is high, DMS emissions decline as the concentration of DMSO increases in the coral holobiont (*Fischer & Jones, 2012*; *Deschaseaux et al., 2014b*). A shift in community structure and decline in the abundance of these species could lead to a significant decline in coral DMS emissions from the GBR. Degraded coral reefs often become dominated by fleshy macroalgae (*Bell, 1992*; *Diaz-Pulido & McCook, 2002*; *De'ath & Fabricius, 2010*; *Barott & Rohwer, 2012*), some of which (e.g., *Polysiphonia* and *Ulva* spp.) are also capable of producing high concentrations of DMSP (*Van Alstyne & Puglisi, 2007*; *Liu et al., 2020*) and may counteract a decline in overall coral reef DMS emissions. Recent work has shown that climate change may result in an increase in seawater DMSP concentration in the tropics, primarily due to an increase in DMSP/O biosynthesis in a range of coral reef taxa, and an increase in the biomass of DMSP-producing fleshy macroalgae (*Green, 2019*). However, the implications for coral reef health and community structure, and whether this may assist coral reefs in coping with ongoing climate change via antioxidant activity or climate regulation, remains highly uncertain.

The ability of corals to adapt to the rapidly changing climate will govern changes to DMS emissions from the GBR. Corals have a close association with a range of microbes and therefore harbor a diverse genome (*Bourne, Morrow & Webster, 2016*) that may facilitate rapid phenotypic change in the coral host (*Torda et al., 2017*). Corals can also enhance their thermo-tolerance by changing their endosymbiont composition via zooxanthellae switching or shuffling (*Bay et al., 2016*). *Acropora* spp. favor Clade D endosymbionts when exposed to thermal stress (*Jones & King, 2015*), which can increase their temperature tolerance by 1.5 °C (*Berkelmans & Van Oppen, 2006*). This may be enough to maintain internal homeostasis in the coral holobiont and protect against mild to moderate marine heatwaves. However, the predicted rate of ocean warming may still exceed the tolerance thresholds of temperature tolerant coral taxa (*van der Zande et al., 2020*). Rapid rises in SST remove gradual "warm-up" periods, which are thought to alleviate temperature shock in corals, helping to mitigate oxidative stress prior to past bleaching events (*Ainsworth et al., 2016*). Temperature-tolerant endosymbionts are typically weaker producers of DMSP under current conditions (*Deschaseaux et al., 2014a*; *Bay et al., 2016*), although tolerance does not always predict DMSP biosynthesis (*Steinke et al., 2011*) and will depend on the rate of future ocean warming.

## CONCLUSIONS

Biogenic emissions of DMS are a significant source of atmospheric sulfur, which in remote marine environments, are an important source of secondary sulfate aerosols. These non-sea salt sulfates play a significant role in the local climate of these remote marine

environments, yet emissions of natural aerosols and their precursors remain one of the largest contributors to our uncertainty in aerosol radiative forcing (*Carslaw et al., 2013*), and ultimately our understanding of what determines the climate sensitivity.

Pristine coral reef dense regions such as the GBR, are particularly strong sources of atmospheric DMS, similar in magnitude to highly productive high-latitude oceans. Corals upregulate the biosynthesis of DMSP and catabolism to DMS in response to physiological stress, with both processes important in maintaining coral homeostasis and promoting resilience to rising ocean temperatures. It is hypothesized that coral reef emissions of DMS increase the formation and condensational growth of marine aerosols and CCN, thereby increasing the brightness, lifetime and cover of low-level marine clouds. Local cloud cover is an important determinant of the spatial extent and severity of coral physiological stress and coral bleaching. Thus, enhanced MBA emissions and LLC cover may establish a negative feedback over coral reefs to mitigate coral physiological stress.

This review has discussed evidence of significant links between coral physiological stress, DMS emissions, aerosol loading and local cloud cover over coral reefs, highlighted by a case study on the GBR, Australia. Given the vast size and relatively remote, pristine location of the GBR, it is possible that the 20 $GgSyr^{-1}$ emitted from the 3,000 individual coral reefs and surrounding lagoon waters significantly influences aerosol and cloud properties in north-eastern Australia. However, there remains substantial uncertainty surrounding the importance of DMS emissions in the properties of the local atmosphere above coral reefs, and what implications ongoing coral reef degradation may have on these complex biogeochemical processes.

## FUTURE DIRECTIONS

The current rate of ocean warming and coral reef degradation increases the urgency at which we must improve our understanding of the importance of DMS in the coral reef radiative climate. Non-linear changes in DMS emissions have been reported in response to thermal and light stress in corals (*Fischer & Jones, 2012*; *Jackson, Gabric & Cropp, 2018*). However, the impacts of ocean warming are being exacerbated by ocean acidification, declining water quality and increased susceptibility to disease, predation and competitive displacement. The synergistic impacts of these co-varying stressors on DMS emissions from coral reefs are largely unknown.

It is possible that rising ocean temperatures will lead to an increase in DMS emissions from coral reefs, although as indicated above, there are limits on the DMS increase associated with the onset of bleaching (*Jackson, Gabric & Cropp, 2018*). This possibility could be examined through a modeling approach akin to those used to project future change in open ocean DMS emissions under warming, eg (*Cameron-Smith et al., 2011*; *Gabric et al., 2013*). This approach is limited by the sparse DMS database in coral reef regions (*Lana et al., 2011*), which currently constrains our ability to derive empirical parametrisatons between DMS water concentration and sea temperature. This may also be complicated if corals are capable of acclimating to rising stressors (*Jurriaans & Hoogenboom, 2020*), or if coral reefs become dominated by more temperature-tolerant

species or zooxanthellae types which do not experience significant oxidative stress under warmed conditions. Interestingly, field surveys of the GBR have demonstrated that concentrations of dissolved DMS decline along a gradient of healthy to disturbed coral reefs (*Jones et al., 2007*). Consequently, DMS sea-air flux will likely be lower for coral reefs that are exposed to multiple synergistic stressors. Although, as noted above, DMS(P) biosynthesis from enhanced algal biomass in degraded coral reef systems may counteract a decline in coralline emissions. These are critical areas for future research and will inform the importance of coral reef emissions of biogenic sulfates in local climate regulation.

The predicted increase in the frequency and severity of mass coral bleaching events may require the implementation of biological and/or physical interventional management strategies. The propagation of temperature-tolerant coral species may allow coral reefs to recover from recent bleaching events (*Van Oppen et al., 2015*). Physically mitigating the warming effects of GHG through solar radiation management (SRM) may also assist corals in coping with future temperature rises. SRM strategies essentially mimic natural biogeophysical processes and involve injecting sea salt aerosol or sulfates into the atmosphere above coral reefs to increase the brightness of LLC (*Crabbe, 2009*; *Latham et al., 2013*; *Irvine et al., 2017*). Several modeled scenarios have found that this significantly reduces the incidence of mass coral bleaching predicted to occur in the GBR, French Polynesia, Caribbean and other tropical coral reefs to the end of this century (*Latham et al., 2013*; *Kwiatkowski et al., 2015*; *Zhang, Jones & Crabbe, 2018*). An additional benefit of these SRM strategies is the potential reduction in the severity of tropical cyclones with a decline in SST. Although climate engineering is a cost and resource-intensive option, it may be necessary to provide short-term protection for high-value or vulnerable coral reefs from rising temperatures.

There is enormous incentive to improve our understanding of the drivers of coral resilience, including the role of dimethylated sulfur compounds in alleviating oxidative stress and influencing the radiative balance. Future research needs to focus on the quantification and characterization of the flux of these compounds from coral reefs and its influence on aerosol and cloud formation. An improved understanding of these biogeophysical processes will provide insight into how to enhance the natural defense mechanisms of corals and inform climate engineering proposals, which may need to be implemented as a last resort to conserve coral reefs in the face of ongoing climate change.

### Funding
This work was funded by a CSIRO PhD top-up scholarship to Rebecca Jackson. There was no additional external funding received for this study. The funders had no role in study design, data collection and analysis, decision to publish, or preparation of the manuscript.

### Grant Disclosures
The following grant information was disclosed by the authors:
CSIRO PhD top-up scholarship.

## Competing Interests

Albert Gabric is an Academic Editor for PeerJ.

## Author Contributions

- Rebecca L. Jackson conceived and designed the experiments, performed the experiments, analyzed the data, prepared figures and/or tables, authored or reviewed drafts of the paper, and approved the final draft.
- Albert J. Gabric conceived and designed the experiments, performed the experiments, analyzed the data, prepared figures and/or tables, authored or reviewed drafts of the paper, and approved the final draft.
- Roger Cropp conceived and designed the experiments, performed the experiments, analyzed the data, authored or reviewed drafts of the paper, and approved the final draft.

## Data Availability

The article is a literature review and has not generated any raw data.

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
