# Peer review of "Coral reefs as a source of climate-active aerosols"

_PeerJ, doi:10.7717/peerj.10023_

## Round 0.1 · original submission · Major Revisions

This manuscript was reviewed by three leaders in the field of marine aerosols, and their recommendations are both clear and extensive. I would expect that in the revision and response to reviewers there is a clear effort to address each suggestion point by point and incorporate these recommendations into the manuscript.

One important note is that you, the authors, have recently published a very similar literature review article in Biogeosciences. There are some clear distinctions, in that this manuscript focuses nominally, narrowly on aerosols, but in reality the two manuscripts have limited differentiation (the biogeosciences paper even has a significant section on marine aerosols which is hardly different) and come close to violating our copyright rules. It is absolutely critical that the authors include in their response a clear explanation of how this manuscript differs from the Biogeosciences paper, including a detailed explanation of modifications you have made to this manuscript in order to differentiate it from the other paper.

Reviewer 1 ·

Basic reporting

No comment

Experimental design

No comment

Validity of the findings

No comment

Additional comments

The review sums up what is known about potential links between DMSP production by coral, aerosol production via nss sulfate oxidation, low level clouds, SST, irradiance, and coral response. Although all of these links are highly uncertain, the authors are able to pull information from enough studies together to create different scenarios based on potential responses of coral to stressors. One concern is the likening of the coral reef – aerosol – climate system to the CLAW hypothesis throughout the paper. If the CLAW hypothesis is to be invoked, evidence against it should also be mentioned including the disconnect between a given coral reef and feedbacks on local climate due to long range transport of sulfate and the time it takes from production of DMSP to the production of nss sulfate and formation of CCN. An additional disconnect is hinted at it Figure 1 – emission of DMS to the marine boundary layer but production of new particles in the free troposphere. This is not to say that coral reefs do not have an impact on climate but rather that a local climate bio-feedback regulation mechanism as suggested by CLAW is unlikely.

Lines 134 – 135: Organo-halogens likely only contribute to new particle formation in coastal regions like Mace Head (e.g., McFiggans, Nature, 433, https://doi.org/10.1038/nature03372, 2005).
Also – please be more specific on “other organic compounds”? What organic compounds other than DMS are of sufficient concentration in the marine atmosphere to be a significant contributor to the particle or CCN number concentration? And does this statement relate to all open ocean regions or only coral reef regions?

Lines 136 – 137: Please explain what is meant by “until recently”. What newly identified sources of MBA have been reported?

Line 229: Should be “…although ocean temperatures NORTHEAST OF Australia…”

It appears that the Jackson et al. 2018a and 2018b papers are the same paper.

Table 1: It is interesting that of the coral reef regions listed, only the GBR stands out with maximum atmospheric DMS concentrations higher than non-coral reef regions. Is this a function of the expanse of GBR?

Future directions: Many of the papers cited are based on modeling studies yet modeling is not discussed in this section. A paragraph on how to best to model this complicated system to predict coral response to stressors would be an important addition.

Reviewer 2 ·

Basic reporting

In “Coral reefs as a source of climate-active aerosols” Jackson, Gabric and Cropp review the impact of coral reefs on local climate through stress induced emissions of aerosol forming precursors.
The review discusses a very interesting topic at the interface of coral reefs as threatened ecosystems and the response of biological changes inside this system that might influences climate through secondary aerosols.
I assume this topic will be very interesting both for coral reef ecologist as well as climate and atmospheric scientist. However, there are several points the author might want to address before the review can be published.

Line 31: Sounds odd. Services to whom? Maybe specify.
Line 39: “and” sounds repetitive. Consider rephrasing this sentence.
Line 43: is this information about alternate states necessary?
Line 72: Maybe specific whether these are secondary aerosols.
Line 108: Its great that the authors provide their literature search methods. Where the here described search terms the only search input? Maybe the Search Term “coral AND aerosol” would have been useful as well. To provide the reader an overview of the efficiency of the search terms maybe the authors could provide the number of hits for the different terms.
Line 122: Consider revising the last sentence about enhancement of the cooling effect of aerosols. According to IPCC Fifth Assessment Report (2013), “earlier assessments considered the radiative implications of aerosol-cloud interactions as two complementary processes—albedo and cloud lifetime effects—that together amplify forcing”, as the authors imply from line 122 to 124. “However, compensating adjustments have been identified that make the system less susceptible to perturbation than might have been expected based on the earlier albedo and lifetime effects. Increases in the aerosol can therefore result in either an increase or a decrease in aerosol–cloud-related forcing depending on the particular environmental conditions.”
Line 135: Maybe expand a little on primary aerosols here. How are they generated and how relevant is this in coral reefs?
Line 182: IS there a study supporting the claim of oxidative stress protection?
Line 272-273: marine biogenic aerosol and continental aerosols come from different sources and have different composition and properties. Consider rephrasing.
Line 278: stress may not be the only cause of DMS emissions. Could differences in primary productivity be a more plausible cause for the seasonal shifts?
Line 287: the previous paragraph indicates that the northern GBR sees the highest aerosol concentration in winter. This sentence seems to disagree with that assumption. The authors may want to distinguish between nucleation events with total aerosol emissions as these processes might not be coupled.
Line 365: Are MBA the only contributors to aerosol optical depth? Even if the link between DMS and aerosol production is sustained, corals are not the only source of DMS. The conclusion in line 368 needs more evidence.

Experimental design

no comment

Validity of the findings

no comment

Additional comments

no comment

·

Basic reporting

The manuscript is written in clear and unambiguous, professional English throughout.
Literature references, sufficient field background/context provided.
The structure of the article is appropriate for a literature review.
Two of the Figures are relevant to the content of the article, and are appropriately described and labeled. The third Figure is not informative but illustrative of the diversity of coral reefs. I would keep it.
There are not raw data to share.

The review is of broad and cross-disciplinary interest and within the scope of the journal.

To my knowledge, the field of coral reef-aerosol-cloud interactions has not been reviewed recently. The review is interesting and timely.

The Introduction is clear and comprehensive, it adequately introduces the subject and makes it clear what the motivation is.

Experimental design

The article content is within the Aims and Scope of the journal.

In my opinion, a rigorous literature search have been performed.

References are relevant and unbiased. Sources are adequately cited.

With a few exceptions (see the general comments to the authors), the review is organized logically into coherent paragraphs and subsections.

Validity of the findings

The observations and arguments reviewed address the major questions posed in the Introduction.

The Conclusions summarize the main points of the manuscript, yet they are a bit disappointing because they essentially consist of a list of unknowns and uncertainties. Honest enough, though. There is a section named Future directions, which is meant to outline where to move towards, although it makes the case for the importance of pursuing this research more than giving concrete directions.

Additional comments

The manuscript reviews current knowledge of the production of aerosol precursors by coral reefs, and their potential to affect the regional radiative budget and alleviate the physiological stress of temperature and solar radiation on corals. The topic is timely because coral reefs are amongst the most diverse ecosystems on Earth, and ones that provide invaluable ecosystem services. There is strong evidence that they are suffering losses and transformations due to global warming, eutrophication and other severe environmental stressors. Understanding how coral reefs cope with stressors is of crucial importance to anticipate their evolution and design conservation policies. The review addresses an atypical and speculative mechanism (I would name it “emergent mechanism” because it involves several environmental compartments with distant connection and unlikely or indirect evolutionary drivers): coral reefs are strong emitters of aerosol-forming substances, mainly DMS, which enhance the formation, lifetime and albedo of low stratiform clouds, hence reducing the solar irradiance on the reef.

The authors mostly focus on the Great Barrier Reefs (GBR) essentially because most relevant data come from these reefs. But, in my opinion, there is another reason: if the proposed mechanism occurs, it is most relevant to the GBR, because it is vast enough for local emissions to see a local effect on clouds. In smaller reefs, any effect on clouds will occur hundreds of km downwind, making very unlikely that these effects will feed back on the emitting ecosystem. As a matter of fact, even the relevance for the GBR must be demonstrated. In my opinion the authors make a strong case that small aerosol numbers, and low-level cloud cover, are enhanced over the GBR. However, some reference is missing on reaction and transport times to support the feedback hypothesis. Lifetime to oxidation of a major aerosol precursor, DMS, in the atmosphere is in the order of 0.5-1 days. What are tyipical transport velocities? Is it feasible that DMS oxidation to sulfuric acid plus nucleation plus growth plus activation as CCN all occur in the air mass still over the GBR?

Specific comments:

I like the introduction, it is succinct but comprehensive enough to set the stage.

L228: Similar tot what? I guess you mean similar to WPWP, but this was two sentences before.

You repeat several times that Acropora spp are the dominant reef-forming corals.

Even though the point is well made that aerosol numbers are enhanced over the GBR with respect to the open ocean, could this also be contributed by increased sea-spray production in the surf zone (I mean, by wave formation and water splashes to the reef)?

L340: exceeds (s missing)

L337-351-368: I think the wording is a bit confusing, and it is not clear under what conditions DMS emissions are maximal. Acclimation to increasing thermal stress favours DMS emission, but severe stress suppresses it. It is better explained further on, but a better organization of the sentences would be appreciated.

You talk about MBA all the time, and sometimes develop them as “DMS and other volatiles”, but you actually focus on DMS. No hint is provided as to what other MBA precursors could be. Is there evidence of any remarkable emission of other MBA?

L390-397: Not clear to me if you buy Green’s arguments. Will the replacement of corals by fleshy algae result in similar/increased/reduced DMS emissions?

L424-426: Rewrite sentence – both what?

Figure 1: how do you connect DMSO to DMS (this way) through photo reactions?
Is DMSP not released by corals? Only through grazing on phytoplankton? Only DMS and DMSO are realeased by corals? It is not totally clear from the photo-diagram who does what. Grazing on phytoplankton is next to the arrow that originates in the coral.

Figure 2: In my opinion, the arrow from nucleated nss-sulfate should connect to marine aerosol, not to CCN. That is: marine aerosol can originate either from (a) condensational growth of pre-existing particles (by pre-existing I mean transported into the reef MBL from elsewhere, or locally emitted as primary aerosol), or (b) nucleation of sulfuric acid and further growth (either in the MBL or by entrainment from the FT). Then, part of the marine aerosol will activate as CCN.

---

## Round 0.2 · Minor Revisions

The reviewers all agreed to review the revised version, and made a few additional recommendations for improvement which you should attend to.

Reviewer 1 ·

Basic reporting

No comment

Experimental design

No comment

Validity of the findings

No comment

Additional comments

The authors have adequately addressed the reviewer comments.

Reviewer 2 ·

Basic reporting

The authors adressed most of my comments, although many of the responses seem a littel spares. Mayeb reconsdier the influience of primary SSA, that are also formed through breaking waves in the surf zone.

Experimental design

NA

Validity of the findings

NA

Additional comments

NA

·

Basic reporting

No comment

Experimental design

No comment

Validity of the findings

No comment

Additional comments

You have adequately addressed my comments and concerns as well as those from the other reviewers.

However, the figures still contain some flaws:

Fig 1 - DMSO is not converted into DMS by photolysis. To my knowledge, this can occur in aqueous medium in the presence of some photosentitizers, but has not been shown in seawater. Rather, DMSO undergoes microbial reduction to DMS (Griebler & Spezak 2000, https://doi.org/10.1080/03680770.1998.11901690; Asher et al. 2017, https://doi.org/10.1002/2016JC012465). Therefore, photolysis should be replaced with microbial reduction.

Fig 1 - An arrow should connect free-living algae to DMS, since not all phytoplankton DMS production occurs through dissolved DMSP upon senescence or grazing, but quite a proportion of the DMS is released or leaked by microalgae, most likely under stress.

Fig 2 - Atmospheric reactions as depicted are a bit misleading. I know it would be impossible (or beyond the scope of this paper) to draw a simple diagram with all the pathways and processes. But the drawing seems to imply that H2SO4 is the species transported to the FT, and it is not shown what happens up there. A number of papers suggest that part of the DMS makes it to the FT, where it is oxidised to H2SO4, which, due to the low condesational sink, nucleates to form new particles that can partly entrain into the MBL. Suggestion: you could delete the arrow from H2SO4 to the FT but send a parallel arrow from DMS to the FT (labeled vertical transfer), then simply write "oxidation and nucleation" up there, and connect it to the entrainment. I think this would be fair enough.

---

## Round 0.3 · accepted · Accept

Thank you for attending to the figure adjustments suggested.